# Design of the Threshold-Controllable Memristor Emulator Based on NDR Characteristics

**DOI:** 10.3390/mi13060829

**Published:** 2022-05-26

**Authors:** Mi Lin, Wenyao Luo, Luping Li, Qi Han, Weifeng Lyu

**Affiliations:** School of Electronics and Information, Hangzhou Dianzi University, Hangzhou 310018, China; luowenyao1226@126.com (W.L.); llp_shadow@163.com (L.L.); handabaojhs@163.com (Q.H.); lvwf@hdu.edu.cn (W.L.)

**Keywords:** memristor emulator, controllable threshold, NDR

## Abstract

Due to the high manufacturing cost of memristors, an equivalent emulator has been employed as one of the mainstream approaches of memristor research. A threshold-type memristor emulator based on negative differential resistance (NDR) characteristics is proposed, with the core part being the R-HBT network composed of transistors. The advantage of the NDR-based memristor emulator is the controllable threshold, where the state of the memristor can be changed by setting the control voltage, which makes the memristor circuit design more flexible. The operation frequency of the memristor emulator is about 250 kHz. The experimental results prove the feasibility and correctness of the threshold-controllable memristor emulator circuit.

## 1. Introduction

Memristors have received widespread attention in the world since HP Labs successfully manufactured TiO_2_ memristors in 2008 [1,2]. A memristor is a non-linear resistor with memory characteristics, and its *i*-*v* characteristic curve is related to the frequency. Its proven good performance implies significant potential in the fields of chaotic circuits [3,4,5], non-volatile memory [6,7], digital logic [8,9], artificial neural networks [10,11,12], and non-linear circuits. In general, the research of memristors includes physical implementation [2,13,14], applications in electronic circuits [15,16], and memristor emulators [17,18,19,20,21,22,23,24,25,26].

The studies on memristor emulators have achieved many results, and a variety of different memristor emulator circuit structures have been proposed, which can be divided into two categories: grounded memristor emulator circuits [18,19] and floating memristor emulator circuits [20,21,22,23,24,25,26]. To design a grounded memristor emulator is easier than a floating memristor emulator. However, due to the fact that one terminal is grounded, the grounded memristor emulator has limited features in circuit designs and applications, and is not suitable for use as a two-terminal device in more complicated circuits [20]. Compared with grounded memristor emulators, the application of floating memristor emulators is more flexible, which can be realized by various active elements, such as second-generation current conveyor (CCII) [21], operational transconductance amplifier (OTA) [22], multi outputs OTA [23], current voltage differencing transconductance amplifier (VDTA) [24], and voltage differencing current conveyor (VDCC) [25] with accompanying passive elements and possibly analog multipliers (AD633) [26]. Although some commercial memristor chips are already available on the market, e.g., Knowm.com (accessed on 9 May 2019) [27], memristor emulators have yet to be widely studied and adopted due to their lower cost.

This article proposes a novel floating threshold-type memristor emulator devised based on the negative differential resistance (NDR) characteristic. Not only does the proposed memristor emulator have a high operating frequency, but it also possesses the controllable threshold voltage, an attribute that makes it more suited for digital logic circuit applications. 

The rest of this paper is organized as follows: in Section 2, the NDR hysteresis unit is introduced and improved on the basis of the R-HBT-NDR unit, and its simulation and hardware experiments are carried out. Section 3 presents the circuit diagram of the threshold-controllable memristor based on NDR characteristics. Section 4 provides the specific circuit implementation and the detailed analysis of its operating principle. In Section 5, the simulation experimental results are presented. Section 6 summarizes the paper.

## 2. Improvement and Design of NDR Hysteresis Unit

Kwang-Jow Gan proposed an R-HBT-NDR unit [28,29,30] with negative resistance characteristics, as shown in Figure 1.

When a sinusoidal signal *V*_AB_ is applied to the R-HBT-NDR unit, its *i*-*v* characteristic curve has a certain hysteresis, as shown in Figure 2 [31].

With the increase in *V*_AB_, transistor *Q*_2_ turns on first, and then transistor *Q*_1_; the current flowing through the R-HBT-NDR unit increases. When *V*_AB_ continues to increase, *Q*_1_ reaches the saturated state, which triggers *Q*_2_ to turn off. The current flowing through the R-HBT-NDR unit then decreases considerably from *I*_P_ to *I*_V_, where negative resistance appears. With the further increase of *V*_AB_, *Q*_2_ remains off, while *Q*_1_ enters into the deep saturation state, and the current flowing through the R-HBT-NDR unit increases slowly with the change of the input signal *V*_AB_. 

When *V*_AB_ starts to decrease, *Q*_1_ enters the cut-off state; *Q*_2_ turns on, and then the current flowing through the R-HBT-NDR unit will increase. As the input signal continues to decrease, *Q*_2_ turns off to end.

Therefore, the movement path of the operation points differs with the increase and decrease of *V*_AB_, resulting in a hysteresis loop. 

Figure 2 only presents the forward part of the curve because of the feature of the NPN transistor. If an extra R-HBT-NDR unit is implemented in the circuit, which consists of PNP bipolar transistors, shown as Figure 3a, under the excitation of the sinusoidal signal, the bidirectional hysteresis curve can be obtained, as shown in Figure 3b.

It can be seen from Figure 3b that the characteristic curve of the bidirectional NDR hysteresis unit shows two hysteresis loops. Meanwhile, the curve has a slope of 0 near the origin, indicating that the change of *V*_AB_ will not affect the resistance, which can be attributed to the influence of the transistor turn-on voltage. Therefore, an improved NDR hysteresis unit is realized, as shown in Figure 4a, and its simplified symbol can be expressed as Figure 4b.

The improved NDR hysteresis unit differentiates its structure from Figure 3a in its two terminals of *V*_+_ and *V*_−_, which have the same voltage magnitude and opposite signs. With the change of *V*_+_ and *V*_−_, the bias voltage of each device in the NDR hysteresis unit will change accordingly, resulting in the different shape of the output curve, as shown in Figure 5. 

In Figure 5a, *V*_+_ = − *V*_−_ = 2 V, it can be seen that when *V*_AB_ = −4.8 V or 4.6 V, the state of the hysteresis curve is changed. In Figure 5b, *V*_+_ = − *V*_−_ = 6 V, when *V*_AB_ = −5.4 V or 5.3 V, the state of the hysteresis curve is changed.

A hardware circuit of the improved NDR hysteresis unit is completed, as shown in Figure 6a. A resistor is connected in series as a load, and the current can be observed by measuring the voltage on the load resistor. Under the excitation of a sinusoidal signal with an amplitude of 6 V and a frequency of 5 kHz, *V*_+_ = *V*_1_ = 6 V, *V*_−_ = *V*_2_ = −6 V, *R*_1_ = *R*_7_ = 10 Ω, *R*_2_ = 240 Ω, *R*_3_ = *R*_5_ = *R*_9_ = *R*_11_ = 2 kΩ, *R*_4_ = *R*_10_ = 8.8 kΩ, *R*_6_ = *R*_12_ = 50 Ω, *R*_8_ = 280 Ω. The experimental results are coincident with the simulation results.

By analyzing the curve in Figure 6, it can be found that it is very similar to the *i*-*v* curve of the TEAM threshold memristor proposed in 2013 [32], shown in Figure 7.

The definition of the threshold type memristor can be expressed as follows [25]:
(1)
$$G=\left\{\begin{array}{l} G_{H}\\ \mathrm{hold}\\ G_{L} \end{array}\right.\begin{array}{c} V_{\mathrm{th}}{}_{1}<V_{X}\\ V_{\mathrm{th}}{}_{2}\leq V_{X}\leq V_{\mathrm{th}}{}_{1}\\ V_{X}<V_{\mathrm{th}}{}_{1} \end{array}$$


*G* is the memductance of the threshold memristor, with *G*_H_ and *G*_L_ corresponding to the two resistance states of the memristor: low resistance (*L*_RS_) and high resistance (*H*_RS_), respectively. *V*_th1_ and *V*_th2_ represent the threshold voltages of the memristor. When the voltage *V*_X_ applied to the memristor is greater than *V*_th1_, the memristor switches to *G*_H_; when the voltage *V*_X_ is less than *V*_th2_, the memristor is switched to *G*_L_; if the voltage applied to the memristor is between *V*_th1_ and *V*_th2_, the state of the memristor remains unchanged.

The hysteresis characteristic curve of the improved NDR hysteresis unit is so similar to the threshold-type memristor that we can utilize this unit to design a novel NDR memristor emulator. This is where the inspiration of this paper originates.

## 3. Circuit Structure Analysis of Threshold-Controllable Memristor Based on NDR Unit

A diagram of the threshold-controllable memristor based on the NDR unit is shown in Figure 8, with an improved bidirectional NDR hysteresis module as the core. It also includes the memductance conversion module, multiplier module, and current transmission module.

The improved NDR hysteresis module U_1_ realizes the hysteresis characteristics; the memductance conversion module U_2_ realizes the non-volatile and high and low resistance conversion characteristics of the memristor; the multiplier module U_3_ converts the current of the memristor into a proportional voltage; the current transmission module U_4_ ensures the equal flow of the current at either end of the memristor.

*V*_ctr,_ represented by *V*_+_ and *V*_−_ in Figure 4, is the control voltage. According to the above analysis, by changing the voltage of *V*_ctr_, the threshold of the memristor will also be changed to realize the change of the memristor resistance state. The symbol of the threshold-controllable memristor emulator can be expressed in a simplified form, as shown in Figure 9.

## 4. Circuit Design of Threshold-Controllable Memristor

The threshold-controllable memristor emulator circuit based on the improved NDR hysteresis unit is shown in Figure 10. The specific analysis of each module is as follows.

### 4.1. Improved NDR Hysteresis Module U_1_

U_1_ is the improved NDR hysteresis module. *V*_ctr_ is a threshold control terminal. When *V*_ctr_ changes, the change trajectories of output *V*_NDR_ are shown in Figure 11. The input voltage is a sinusoidal signal with an amplitude of 6 V and a frequency of 5 kHz.

A relationship is found between *V*_ctr_ and *V*_NDR_: when *V*_ctr_ increases, *V*_NDR_ will increase accordingly, which can be expressed as:
(2)
$$V_{\mathrm{NDR}}=f(V_{\mathrm{ctr}})$$


### 4.2. Memductance Conversion Module U_2_

U_2_ is the memductance conversion module, composed of a four-channel operational amplifier TL084, a capacitor, and several resistors, which is used to achieve the non-volatile and high/low resistance characteristics of memristors. The output *V*_NDR_ of U_1_ is the input signal of U_2_.

Since the memristor has the non-volatile characteristic, its resistance must be related to the previous state. *R*_mc_ and *C*_mc_ form the integrator with the output *V*_nv_ of:
(3)
$$V_{\mathrm{nv}}=-\frac{1}{R_{\mathrm{mc}}C_{\mathrm{mc}}}\int V_{\mathrm{NDR}}dt=-\frac{1}{R_{\mathrm{mc}}C_{\mathrm{mc}}}\int f(V_{\mathrm{ctr}})dt$$


Figure 12 presents the characteristics of the bistable circuit that is composed of part B of TL084.

In Figure 12, when *V*_nv_ is greater than *V*_TH_, the output is −*V*_CC_; when *V*_nv_ is less than *V*_TL_, the output is *V*_CC_; when *V*_nv_ is between *V*_TL_ and *V*_TH_, the state will not change. −*V*_CC_ and *V*_CC_ represent the power supply voltages of TL084. Therefore, the output voltage *V*_BC_ of the bistable circuit is expressed as follows:
(4)
$$V_{\mathrm{BC}}=\left\{\begin{array}{l} -V_{\mathrm{CC}}\\ \mathrm{hold}\\ V_{\mathrm{CC}} \end{array}\right.\begin{array}{c} V_{\mathrm{TH}}<V_{\mathrm{nv}}\\ V_{\mathrm{TL}}\leq V_{\mathrm{nv}}\leq V_{\mathrm{TH}}\\ V_{\mathrm{nv}}<V_{\mathrm{TL}} \end{array}$$

and the threshold voltages of *V*_TH_ and *V*_TL_ are expressed as follows:
(5)
$$V_{\mathrm{TH}}=-V_{\mathrm{TL}}=\frac{R_{A}}{R_{B}+R_{A}}V_{\mathrm{CC}}$$


In addition, in order to achieve the single polarity for the output voltage of *V*_mc_, a summing circuit is constructed, which consists of part C of TL084, *V*_3_, *R*_14_, *R*_15_, *R*_16_, *R*_17_, and *R*_18_. The corresponding output *V*_mc_ is expressed as follows:
(6)
$$\begin{array}{l} V_{\mathrm{mc}}=\left\{\begin{array}{l} -(bV_{3}-aV_{\mathrm{CC}})\\ \mathrm{hold}\\ -(aV_{\mathrm{CC}}+bV_{3}) \end{array}\right.\begin{array}{c} V_{\mathrm{TH}}<V_{\mathrm{nv}}\\ V_{\mathrm{TL}}\leq V_{\mathrm{nv}}\leq V_{\mathrm{TH}}\\ V_{\mathrm{nv}}<V_{\mathrm{TL}} \end{array}\\ \;=\left\{\begin{array}{l} -(bV_{3}-aV_{\mathrm{CC}})V_{\mathrm{TH}}<-\frac{1}{R_{\mathrm{mc}}C_{\mathrm{mc}}}\int f(V_{\mathrm{ctr}})dt\\ \mathrm{hold}V_{\mathrm{TL}}\leq -\frac{1}{R_{\mathrm{mc}}C_{\mathrm{mc}}}\int f(V_{\mathrm{ctr}})dt\leq V_{\mathrm{TH}}\\ -(aV_{\mathrm{CC}}+bV_{3})-\frac{1}{R_{\mathrm{mc}}C_{\mathrm{mc}}}\int f(V_{\mathrm{ctr}})dt<V_{\mathrm{TL}} \end{array}\right. \end{array}$$

where *a* = *R*_16_/*R*_14_, *b* = *R*_16_/*R*_15_; compared to the definition of the threshold-type memristor in (1), (*aV*_CC_ + *bV*_3_) can be seen as a voltage representation of the high-memductance state of memristor; (*bV*_3_ − *aV*_CC_) can be seen as a voltage representation of the low-memductance state of the memristor.

### 4.3. Multiplier Module U_3_

U_3_ is the multiplier module and it is composed of a multiplier AD633, which converts the current of the memristor into a proportional voltage. The output of U_3_ can be expressed as follows:
(7)
$$V_{\mathrm{mul}}=\frac{R_{19}+R_{20}}{10R_{19}}V_{\mathrm{NDR}}V_{\mathrm{mc}}=\frac{R_{19}+R_{20}}{10R_{19}}f(V_{\mathrm{ctr}})V_{\mathrm{mc}}$$

where *V*_NDR_ is the output of U_1_ and *V*_mc_ is the output of U_2_. *R*_19_ and *R*_20_ adjust the multiplication coefficient.

### 4.4. Current Transmission Module U_4_

U_4_ is the current transmission module and composed of two AD844 operational amplifiers. This module converts the voltage *V*_mul_ into the corresponding current and ensures equal currents flowing through A and B terminals. The current through the memristor is expressed as follows:
(8)
$$i=\frac{V_{\mathrm{mul}}}{R_{\mathrm{AB}}}$$

and the current flowing through *R*_AB_ is *i*_1_ = *V*_AB_/*R*_AB_; therefore, the memductance *G* can be represented by the voltage *V*_mc_, as follows:
(9)
$$G=\frac{i}{V_{\mathrm{AB}}}=\frac{\left(R_{19}+R_{20}\right)}{10R_{19}i_{1}R_{\mathrm{AB}}^{2}}V_{\mathrm{NDR}}V_{\mathrm{mc}}=\frac{\left(R_{19}+R_{20}\right)}{10R_{19}i_{1}R_{\mathrm{AB}}^{2}}f(V_{\mathrm{ctr}})V_{\mathrm{mc}}$$


Assuming *k*_G_ = (*R*_19_ + *R*_20_)/(10*R*_19_*i*_1_
$R_{AB}^{2}$
) = (*R*_19_ + *R*_20_)/(10*R*_19_*i*_1_
$R_{AB}^{2}$
), *G* can also be expressed as:
(10)
$$G=\frac{i}{V_{\mathrm{AB}}}=k_{G}V_{\mathrm{NDR}}V_{\mathrm{mc}}=k_{G}f(V_{\mathrm{ctr}})V_{\mathrm{mc}}$$


Therefore, the mathematical emulator of the memristor emulator circuit can be expressed as follows:
(11)
$$\begin{array}{l} G=\left\{\begin{array}{l} -k_{G}\;f(V_{\mathrm{ctr}})(aV_{\mathrm{CC}}+bV_{3})\\ \mathrm{hold}\\ -k_{G}\;f(V_{\mathrm{ctr}})(bV_{3}-aV_{\mathrm{CC}}) \end{array}\right.\begin{array}{c} -R_{\mathrm{mc}}C_{\mathrm{mc}}V_{\mathrm{TL}}<V_{\mathrm{nv}}\\ -R_{\mathrm{mc}}C_{\mathrm{mc}}V_{\mathrm{TH}}\leq V_{\mathrm{nv}}\leq -R_{\mathrm{mc}}C_{\mathrm{mc}}V_{\mathrm{TL}}\\ V_{\mathrm{nv}}<-R_{\mathrm{mc}}C_{\mathrm{mc}}V_{\mathrm{TH}} \end{array}\\ \;=\left\{\begin{array}{l} -k_{G}\;f(V_{\mathrm{ctr}})(aV_{\mathrm{CC}}+bV_{3})-R_{\mathrm{mc}}C_{\mathrm{mc}}V_{\mathrm{TL}}<-\frac{1}{R_{\mathrm{mc}}C_{\mathrm{mc}}}\int f(V_{\mathrm{ctr}})dt\\ \mathrm{hold}\;-R_{\mathrm{mc}}C_{\mathrm{mc}}V_{\mathrm{TH}}\leq -\frac{1}{R_{\mathrm{mc}}C_{\mathrm{mc}}}\int f(V_{\mathrm{ctr}})dt\leq -R_{\mathrm{mc}}C_{\mathrm{mc}}V_{\mathrm{TL}}\\ -k_{G}\;f(V_{\mathrm{ctr}})(bV_{3}-aV_{\mathrm{CC}})-\frac{1}{R_{\mathrm{mc}}C_{\mathrm{mc}}}\int f(V_{\mathrm{ctr}})dt<-R_{\mathrm{mc}}C_{\mathrm{mc}}V_{\mathrm{TH}} \end{array}\right. \end{array}$$


Compared with Equation (1), *G* in Equation (11) satisfies the definition of threshold memristors. In addition, when the value of *V*_ctr_ changes, *G* changes, with the threshold-controllable function thereby being achieved.

## 5. Verification of Threshold-Controllable Memristor

### 5.1. Simulation Results

The simulation results of the threshold-controllable memristor are shown in Figure 13: *R*_1_ = *R*_7_ = 19 Ω, *R*_2_ = 7.5 Ω, *R*_3_ = *R*_5_ = *R*_9_ = *R*_11_ = 2 kΩ, *R*_4_ = *R*_10_ = 8 kΩ, *R*_6_ = *R*_12_ = 51 Ω, *R*_8_ = 29 Ω, *R*_13_ = 1 Ω, *R*_14_ = 9.1 kΩ, *R*_15_ = 1 kΩ, *R*_16_ = 10 kΩ, *R*_17_ = *R*_18_ = 1 kΩ, *R*_19_ = 10 kΩ, *R*_20_ = 5.6 kΩ, *R*_mc_ = 10 kΩ, *C*_mc_ = 1 nF, *R*_A_ = 0.5 kΩ, *R*_B_ = 2.5 kΩ, *R*_AB_ = 100 kΩ. The excitation signal is a sinusoidal signal with a 5 V amplitude 30 kHz frequency.

When *V*_ctr_ = 6 V, the threshold voltage of the memristor is about ±2.9 V; when *V*_ctr_ = 2 V, the threshold voltage is −2.6 V and 2.7 V. Therefore, when *V*_ctr_ changes, the memristor threshold voltage changes accordingly. This feature can be applied to digital logic circuits design where the state of the memristor can switch to high or low in order for varied logic states by only changing *V*_ctr_, and different functions will be achieved without modifying the circuit structure and input signals.

### 5.2. Experimental Results

The hardware circuit of the threshold-controllable memristor emulator based on NDR characteristics and its experimental results is shown in Figure 14.

Under the excitation of a sinusoidal signal with an amplitude of 6 V and a frequency of 72 kHz, when *V*_ctr_ = 2 V, its threshold voltages are −2 V and 2.1 V, as shown in Figure 14b. When *V*_ctr_ = 6 V, its threshold is about −2.2 V and 2.8 V, as in Figure 14c. If the frequency keeps increasing until 345 kHz, the hysteresis loop will be missing and the non-volatile characteristic disappears, as in Figure 14e.

## 6. Conclusions

A threshold-controllable memristor emulator based on NDR characteristics is proposed, which is composed of an improved NDR hysteresis unit, multipliers, and operational amplifiers. In addition to the basic characteristics of memristors such as being non-volatile and non-linear, this emulator circuit also has the advantage of threshold controllability and high operating frequency. Compared to the two-terminal structure of the traditional memristor, the threshold-controllable memristor emulator is more suitable for digital circuit design thanks to its additional control terminal structure, where the memristor threshold switches under different control voltages, indicating more flexible and convenient memristor-based circuit designs. Our follow-up effort will be committed to continuously optimizing the performance of the emulator, and the attempt to apply it to the design of memristive digital logic circuits.

## Figures and Tables

**Figure 1 micromachines-13-00829-f001:**
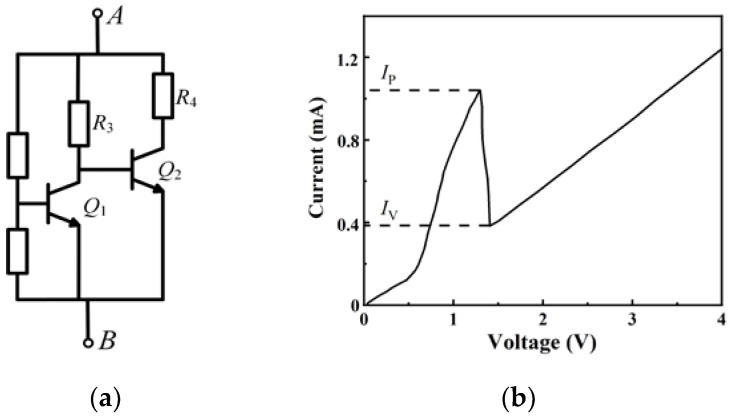
R-HBT-NDR unit and its *i*-*v* characteristic curve: (**a**) R-HBT-NDR unit, (**b**) the *i*-*v* characteristic curve of R-HBT-NDR unit.

**Figure 2 micromachines-13-00829-f002:**
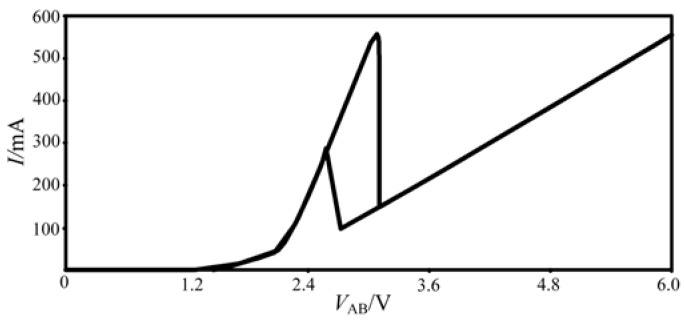
Hysteresis characteristic of the R-HBT-NDR unit.

**Figure 3 micromachines-13-00829-f003:**
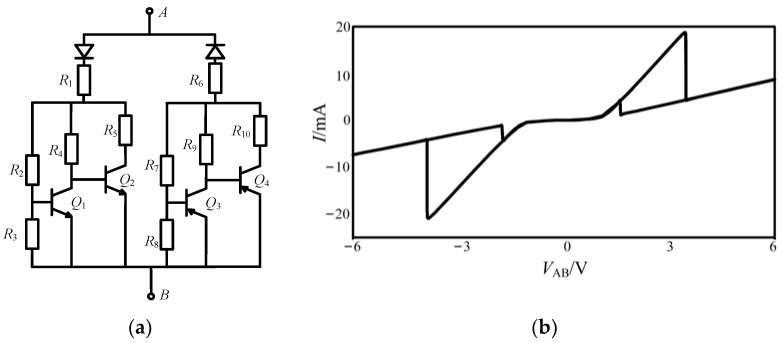
Bidirectional NDR hysteresis unit and its *i*-*v* characteristic curve: (**a**) bidirectional NDR hysteresis unit, (**b**) the *i*-*v* characteristic curve of bidirectional NDR hysteresis unit.

**Figure 4 micromachines-13-00829-f004:**
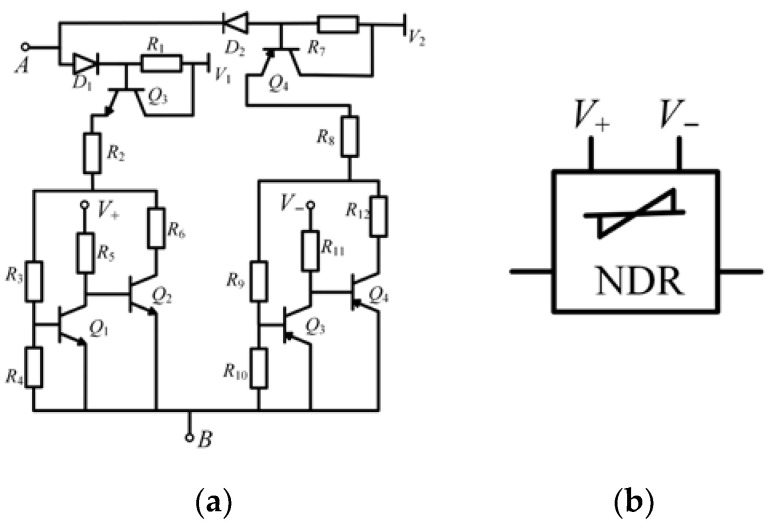
Improved NDR hysteresis circuit and its simplified symbol: (**a**) improved NDR hysteresis circuit, (**b**) the simplified symbol of improved NDR hysteresis circuit.

**Figure 5 micromachines-13-00829-f005:**
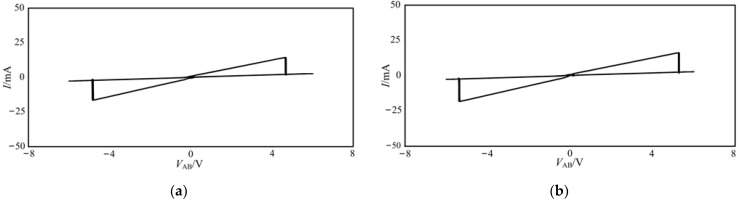
The *i*-*v* characteristic curve of the improved hysteresis circuit: (**a**) when *V*_+_ = − *V*_−_ = 2 V, (**b**) when *V*_+_ = − *V*_−_ = 6 V.

**Figure 6 micromachines-13-00829-f006:**
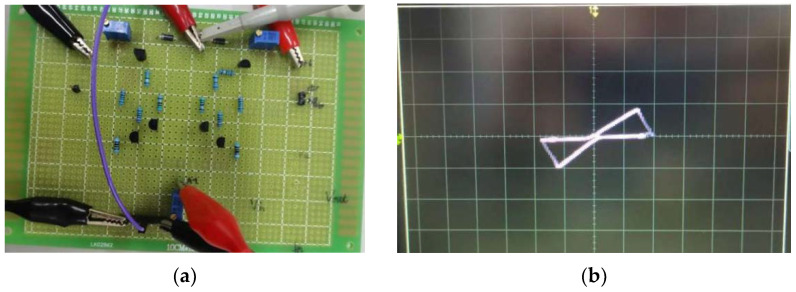
Improved NDR hysteresis unit hardware circuit and its experimental result: (**a**) improved NDR hysteresis unit hardware circuit, (**b**) experimental result.

**Figure 7 micromachines-13-00829-f007:**
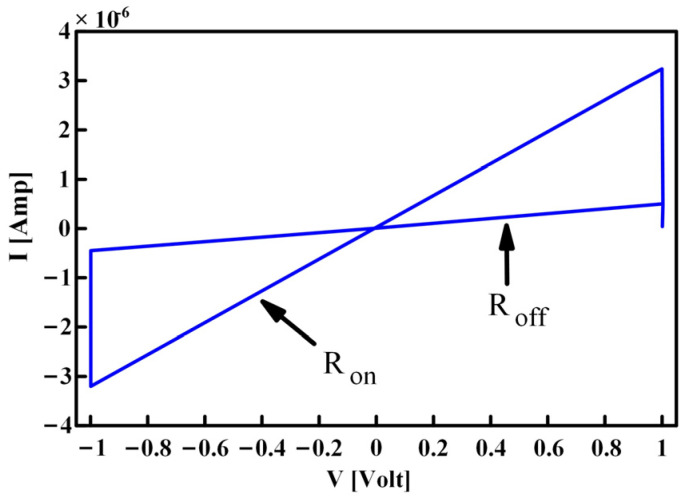
The *i*-*v* characteristic curve of the TEAM threshold type memristor.

**Figure 8 micromachines-13-00829-f008:**
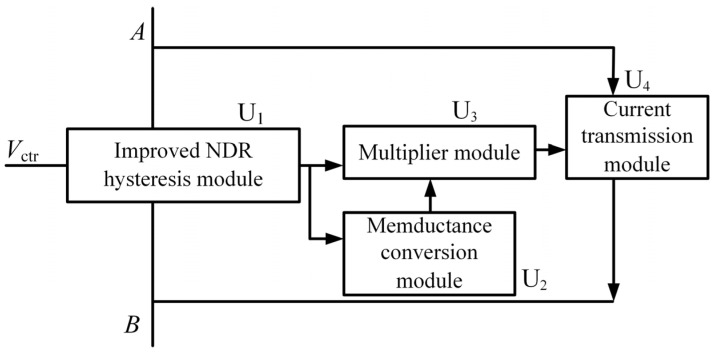
Diagram of threshold-controllable memristor emulator based on NDR units.

**Figure 9 micromachines-13-00829-f009:**
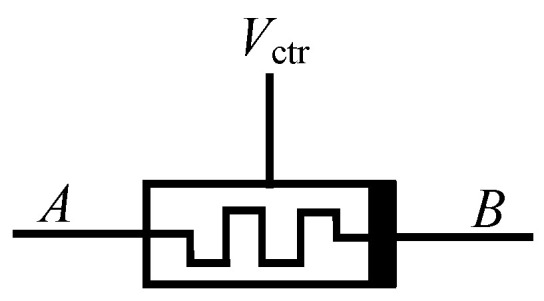
Simplified symbol of threshold-controllable memristor.

**Figure 10 micromachines-13-00829-f010:**
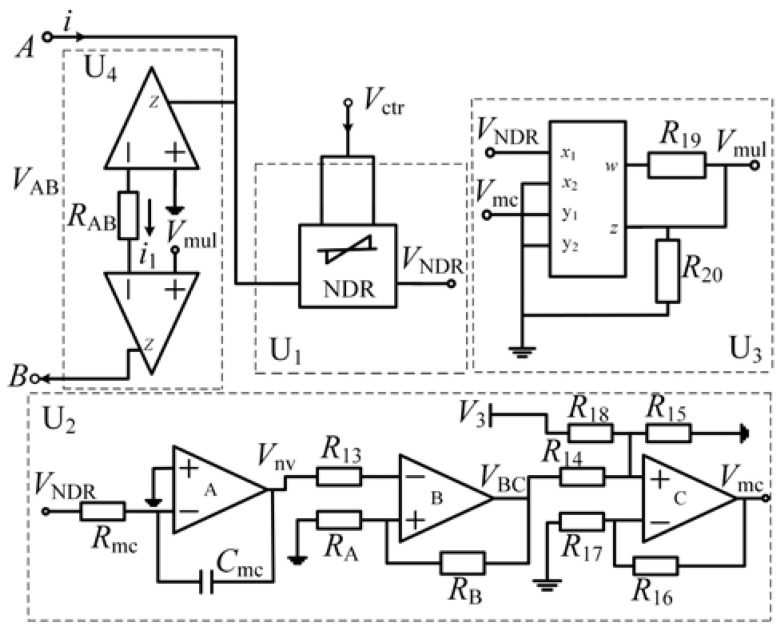
Threshold-controllable memristor emulator circuit based on the improved NDR hysteresis unit.

**Figure 11 micromachines-13-00829-f011:**
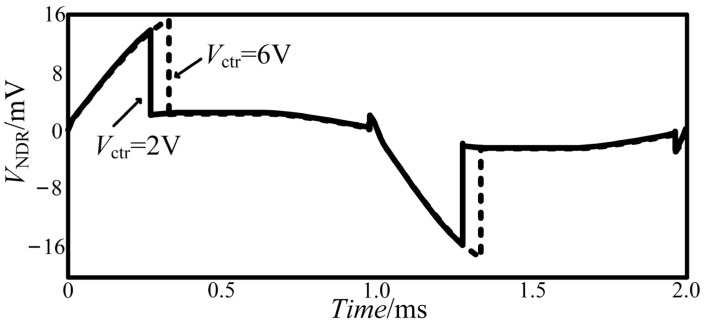
Output curves of *V*_NDR_.

**Figure 12 micromachines-13-00829-f012:**
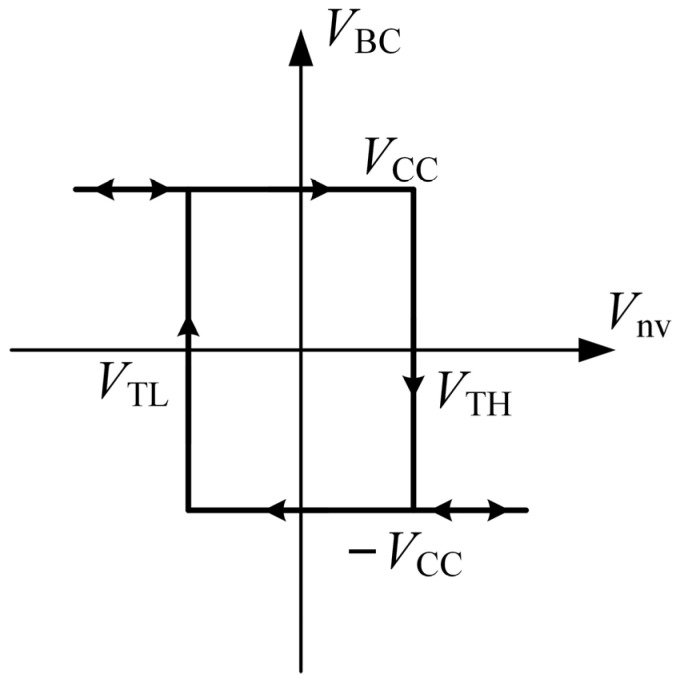
Characteristics of the bistable circuit composed of part B of TL084.

**Figure 13 micromachines-13-00829-f013:**
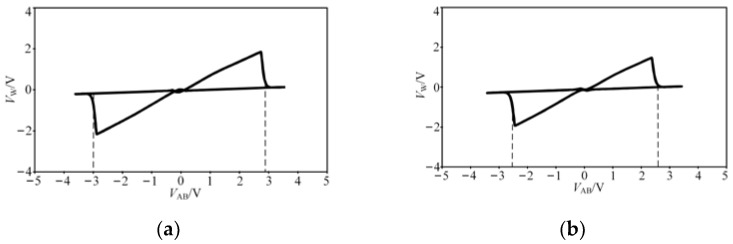
Simulation curves: (**a**) *V*_ctr_ = 6 V, (**b**) *V*_ctr_ = 2 V.

**Figure 14 micromachines-13-00829-f014:**
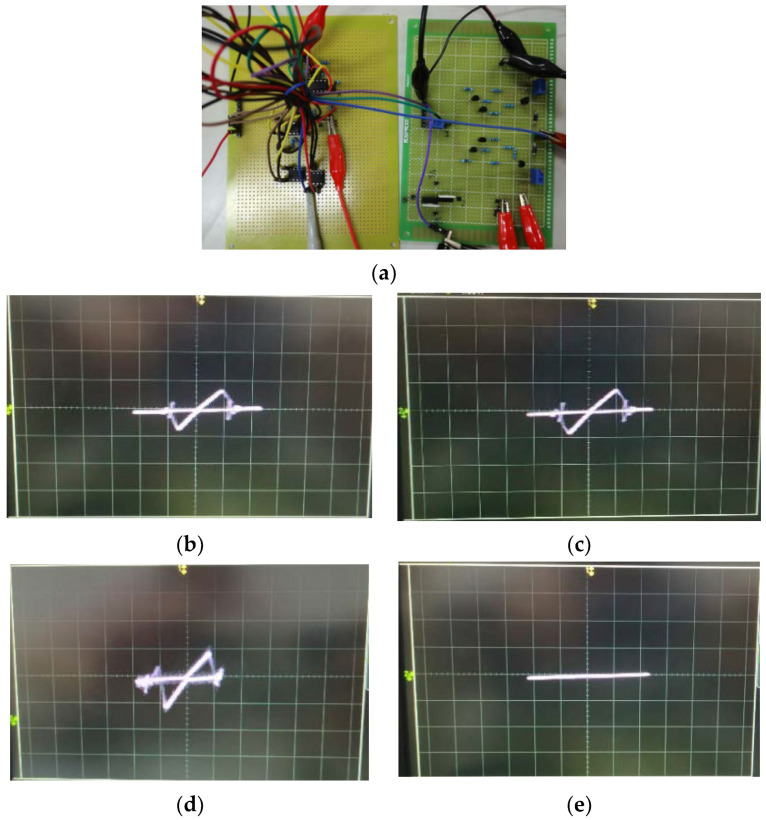
Hardware circuit and experimental results: (**a**) the circuit of the threshold-controllable memristor emulator, (**b**) *V*_ctr_ = 2 V, *f* = 72 kHz, (**c**) *V*_ctr_ = 6 V, *f* = 72 kHz, (**d**) *V*_ctr_ = 6 V, *f* = 250 kHz, (**e**) *V*_ctr_ = 6 V, *f* = 345 kHz.

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
