# Peer review of "Design of the Threshold-Controllable Memristor Emulator Based on NDR Characteristics"

_micromachines, 2022, doi:10.3390/mi13060829_

Round 1

Reviewer 1 Report

The authors have presented a threshold controllable memristor emulator based on NDR characteristics that is composed of an improved NDR hysteresis unit, multipliers, and operational amplifiers. The paper is well written.  But, the quality of the figures must be improved.

Author Response

The pictures in the article have been modified.

Reviewer 2 Report

The work offers an emulation circuits based on the application of NDR as a nonlinear element that provides the desired transfer function. However, it is based on the application of bipolar technology, with many building elements, and a very limited frequency band, which is certainly a lack of practical application and implementation in the form of an integrated circuit. Analysis of present non-linearity’s, nor comparison with similar solutions, was not performed.

Reviewer 3 Report

This paper present the circuit design using threshold switching and negative differential resistance behaviors of memristor. Some meaningful simulation data was conducted.

1) Introduction is very short. Can be discussed more about memristor's application and progress.

2) I cannot clearly see the application of this work. It is not clear which memristors this circuit is applicable to.

3) The memristor I-V curve seems to have been set arbitrarily and may differ from the actual measurement curve. This needs further explanation.

4) Why was BJT used in circuit instead of MOSFET?

Round 2

Reviewer 2 Report

The authors did not respond in a consistent manner or one of the remarks of the previous version of work.

Author Response

The description of the memristor emulator circuit has been revised, the different types of grounded and floating memristor emulators are introduced, and the latest references in the past three years are updated. We have carefully checked and improved the English writing in the revised manuscript.

The threshold controllable memristor emulator based on NDR characteristics proposed in this paper can be applied to the digital logic circuits. When Vctr changes, the memristor threshold voltage changes accordingly. This feature can be applied to digital logic circuits design  where the state of memristor can switch for high or low in order for varied logic states by only changing Vctr and different func-tions will be achieved without modifying the circuit structure and input signals.

Reviewer 3 Report

I think the revised version can be accepted.

Author Response

We have carefully checked and improved the English writing in the revised manuscript.